# Green Production Management and Innovation Nexus: Evidence from Technology-Based SMEs of China

**Lianjie Zhou * and Yuhui Dai**

Entrepreneurship College, Zhejiang University of Finance and Economics Dongfang College, Haining 314408, China
* Correspondence: lizy2006@yeah.net

**Abstract:** This paper aims to study the relationship between green production management and enterprise innovation through empirical analysis of China's technology-based small and medium-sized enterprises (SMEs). It can promote the improvement of the production management efficiency of enterprises. The rapid development of information technology and the change in social productivity has changed lifestyles in ways that trigger certain challenges in production management, especially in technology-based SMEs. The main issue is the role of leaders and organizational practices. Therefore, this paper designs and improves the structural and operating mechanisms of technology-based SMEs by employing the person fit and evolutionary game models. This paper gathers data from technology-based SMEs of Zhejiang Province, China, by conducting a questionnaire-based survey. The principle of person-environment fit revealed the positive leadership skills of enterprise managers. In addition, the evolutionary game model revealed the re-optimization of SMEs to improve management efficiency through reforming enterprises' organization, management, and supervision mechanisms. Finally, strengthening collaborative innovation, improving innovation support services, grasping the balanced scale of the system, and boosting the innovation habitat for the healthy and innovative ecosystem of technology-based SMEs are proposed. This paper provides suggestions for policymakers to expand and upgrade management, especially in technology-based enterprises.

**Keywords:** person-environment fit; technology-based SMEs; evolutionary game model; green production management; innovation structure





## 1. Introduction

With the rapid development of the social economy, innovation has become a key element for competitive, sustainable, and healthy enterprise development. In the contemporary era, green, sustainable, and healthy enterprise development is crucial, and according to the 19th National Congress of the Communist Party in the context of China, it is pointed out that "China's economy had shifted from a stage of high-speed growth to a stage of high-quality development". The marginal role of traditional production factors in China has gradually declined, which fosters the need for innovation to boost economic growth [1]. Small and medium-sized enterprises (SMEs) are the driving force of the national innovation industry and environmental sustainability. With the continuous transition of the economic system to a stage of high-quality development, the total investment of the economy has increased significantly [2]. Weak economic growth, enterprise vitality, and technological achievements have become the main problems of the current industrial structure. Therefore, in recent years, promoting the structural reform of SMEs [3,4] and improving environmental production efficiency have become the main problems of current research.

In technology-based SMEs, the green production management and innovation system is a network structure comprised of multiple features. Under various agglomerations, it can produce nonlinear effects and couple the industrial chain, value chain, and knowledge

chain [5,6]. In the prevailing literature, many scholars have explored the role of green production management in technology-based SMEs. In this vein, Xu et al. [7] designed a financial choice model for technology-based SMEs under the constraints of a low-carbon economy. They analyzed the problems in financial choices and proposed the support of financial mechanisms necessary to develop a low-carbon economy. Likewise, a recent study by Incekara [8] analyzed the impact of various external financial variables on SMEs' adoption of green management. Using the binary logistic regression, the findings revealed that green banking could promote the re-planning of the adoption of water resources and also minimize waste practices. The results showed that peer-to-peer lending was positively and significantly associated with waste reduction. Based on this, it is apparent that enterprise innovation and green production management can boost enterprise development more successfully, especially in the current aggravated environment. However, there is still space, such as low performance and optimizing the entrepreneurial environment, which must be resolved to further promote the development of technology-based SMEs.

It is important to understand the development status and bottlenecks of technology-based SMEs, the dependence and evaluation of high-tech enterprises on the innovation environment, the development status of government public technology service platforms, the transformation rate of scientific and technological achievements and existing problems, and the incubation situation of high-tech entrepreneurial center enterprises and their existing problems. Therefore, technology-based SMEs' green production management and innovation must be investigated and analyzed. In the case of China, technology-based SMEs also face challenges, such as strengthening innovation capabilities, optimizing the entrepreneurial environment, improving the service system, and expanding financing channels. To address these issues and propose certain policy implications, especially in the context of China's technology-based SMEs, this paper integrates the digital psychology and the evolutionary game model to design and enhance the structural model and operation mechanism of the green ecological innovation production system to facilitate the establishment of sustainable and innovative technology-based SMEs. In addition, technology-based SMEs are taken as the research objects to conduct investigation and analysis to provide an experimental reference for the healthy operation and improvement of the innovation ecosystem of technology-based SMEs.

This paper uses evolutionary game theory with the aim to reflect the premise of bounded rationality and reach a game equilibrium state in an individual trial-and-error approach [9]. In the green production management and innovation alliance of technology-based SMEs, members lack sufficient trust at the beginning of the partnership due to the diversity and uncertainty of the internal and external environments. With time, the trust between members improves; consequently, the various alliance systems also improve. Collaborative innovation is finally obtained after repeated games and constant adjustment of the members' individual choices that lead to the stable operation and development of the partnership [10,11]. In addition, the purpose is to optimize the green management and production innovation strategies of technology-based SMEs and further improve the collaboration efficiency of enterprise management. It can promote the restructuring of the industrial structure. At present, there are certain deficiencies in the production management of technology-based SMEs. This paper fills the research gap in the production management of technology enterprises by expanding the theoretical knowledge of production management of technology-based SMEs.

Here, the person-environment fit model is analyzed. In the green production management of technology-based SMEs, the degree of matching between individual and environmental characteristics is likely to subtly influence managers' desires. In a work environment with a high degree of person-environment fit, the talents of enterprise managers can be effectively utilized. Through the principle of consistent matching and the direction of complementary matching, the behavior of individuals depends on the combined roles of both themselves and the environment. In the organizational socialization of the enterprise working environment, individuals and enterprises gradually integrate into the organiza-

tion. Individuals improve interpersonal relationships between team members by adapting and matching an individual's personality to the professional environment, enhancing enterprise management's collaborative efficiency. In addition, appropriate adjustments to the person-environment fit principle in the model require the investigation of the leader's leadership role in organizational practice.

Moreover, the optimization of the production management function of the enterprise can be promoted through the organizational skill division and practical skills training of leaders. In the entrepreneurial practice of enterprises, the important role of leaders is to expand the network between social organizations. Differences in social capital represent differences in an individual's ability to access resources through networks. For the resources that can be transported and stored in the relationship network in enterprise management, it is necessary to train the entrepreneurial spirit and entrepreneurial skills with the help of enterprise leaders and the guidance of social organization workers. Leaders' management skills must be exerted in organizational practice through knowledge and visits. Given the problem of insufficient entrepreneurial ability of enterprises, leaders need to give full potential to their property rights distribution function in organizational practice. The main problem of current research is that the efficiency of enterprise production management and the management mechanism are not matched. Therefore, under the current institutional pattern of the market economy, it is necessary to combine new production strategies of SMEs. The goal is to promote the improvement of the efficiency of enterprise production management. The main contribution is to realize the innovation of the green production management mode through institutional reform.

The remaining parts of this paper are structured as follows. Section 2 covers the theoretical framework. The methodology is explained in Section 3; the data sources and analytical strategies are also presented in this section. The results based on estimations are presented and discussed in Section 4. Finally, a conclusion with certain possible policy recommendations is revealed in the Section 5.

## 2. Theoretical Framework

### 2.1. Green Production and Innovative Management of Technology-Based SMEs

The main scientific knowledge of green production and enterprise management system innovation is included here to analyze the relationship between the production management strategy and innovation management of enterprises. In the development of technology-based SMEs, there are many kinds of technical services in the market [12]. For example, they provide support services for large enterprises and become a core enterprise with technological advantages in a certain segment. In general, the construction of green production and the innovation of ecological management systems in technology-based SMEs are mainly composed of the dynamic mechanism, knowledge sharing mechanism, interest coordination mechanism, and external governance mechanism [13,14]. A theoretical framework capturing this is shown in Figure 1. In Figure 1, firstly, technology-based SMEs have a strong driving force for innovation and actively cooperate with various heterogeneous innovation entities outside the enterprise boundary [15]. They are the foundation and driving force for green production management and the construction of the innovation ecosystem. Secondly, the exchange, integration, motivation, and updating of relevant knowledge are the keys to realizing the value of a technology-based enterprise in building a green production management system. Thirdly, there is a mutually beneficial relationship between the interest coordination mechanism and green production management system in technology-based SMEs.

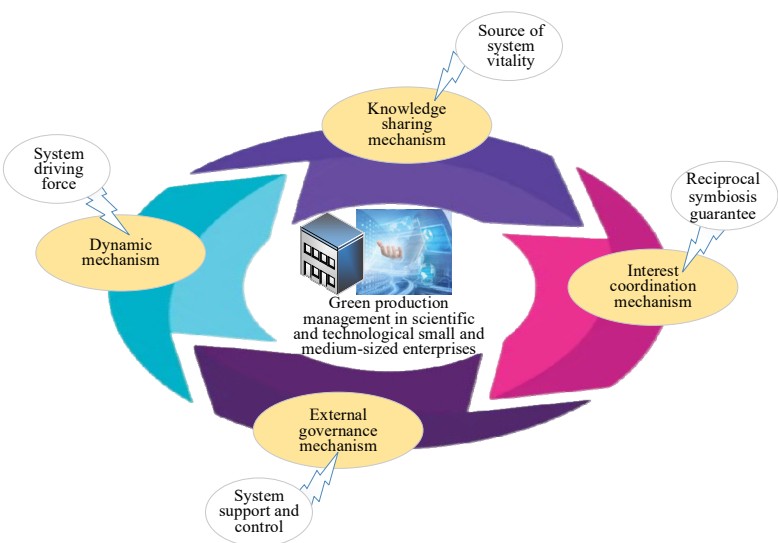

**Figure 1.** Schematic diagram of the theoretical framework of green production and innovative ecological management mechanism in technology-based SMEs.

### 2.2. Core Elements of the Green Production Management and Innovation System of Technology-Based SMEs

Moreover, the core elements of the green production management and innovation system of technology-based SMEs are portrayed in Figure 2. Figure 2 mainly includes the innovation base layer, the innovation body layer, the innovation collaboration layer, the innovation support layer, and the innovation carrier and the environment. The mass innovation source is the innovation base layer and foundation, which refers to groups with certain innovation consciousness and abilities, such as intellectuals, scientific researchers, international students, college students, and ordinary people. The innovation body layer consists of the technology-based SME community; the upper, middle, and lower layers of the technology-based SME industry chain; and competitors [16]. The innovation collaboration layer includes colleges and universities, scientific research institutes, large-scale open laboratories, and the innovation application layer. Different roles and various functions in constructing the green production management system are assumed in this layer [17].

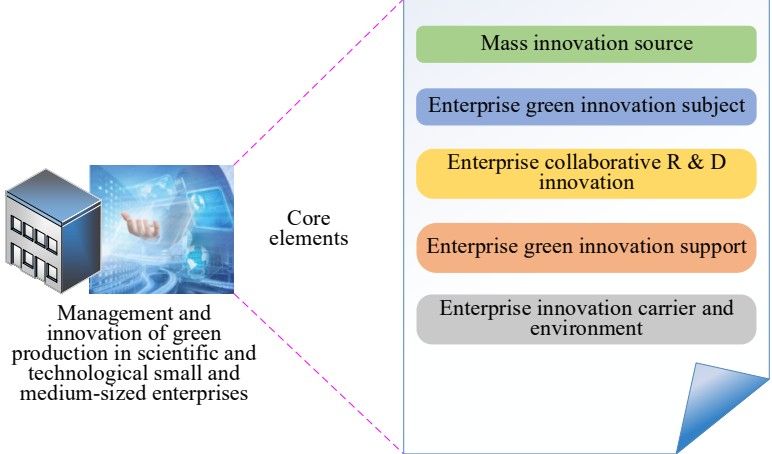

**Figure 2.** Schematic diagram of the core elements of the green production management and innovation system of technology-based SMEs.

The innovation support layer mainly includes the government, technology intermediaries, financial institutions, venture capital institutions, and other social resources or organizations. The innovation support layer provides systematic material support

for technology-based SMEs' green production management and innovation system construction. The innovation carriers of technology-based SMEs mainly refer to science and technology industrial parks and high-tech development zones. These parks and development zones are the carriers of technology-based SMEs and the environmental support for their survival [18]. When the core elements are complete, technology-based SMEs' green production management and evolution can develop in an all-around and market-oriented manner.

## 3. Materials and Methods

### 3.1. Data Sources

The data survey was collected from technology-based SMEs of the Zhejiang Province of China. The survey used in this study gathered information by conducting a face-to-face questionnaire. The questionnaire includes basic information about the enterprise, the internal green production management and innovation of the enterprise, the cooperation between the enterprise and the external innovation, the government agency's support for the enterprise's green production management and innovation, and the environmental factors restricting the enterprise's green production management and innovation. The samples followed certain criteria to ensure credibility and validity. The sample comprises middle- and high-level cadres or front-line scientific and technological personnel of technology-based SMEs. Furthermore, the surveyed enterprises are informed that the data are only used for academic research. The questionnaire structure of the data survey design is shown in Appendix A: Table A1.

A total of 131 questionnaires were distributed, and 124 were recovered, with a recovery rate of 94.66%. There are 119 valid questionnaires, and the effective rate of the questionnaires is 95.97% after the questionnaires with obvious errors and missing questions were eliminated. The statistics of different questionnaire items were counted, and the results are revealed in Table 1.

**Table 1.** Questionnaire statistical results: scoring statistics for different items.

| | Number of Questionnaires | Minimum Value | Maximum Value | Mean Value | Standard Deviation |
|---|---|---|---|---|---|
| Enterprise management | 119 | 2.01 | 6.89 | 4.505 | 1.384 |
| Knowledge sharing | 119 | 1.56 | 6.89 | 3.888 | 1.524 |
| Employee innovation | 119 | 2.69 | 6.89 | 4.194 | 1.628 |
| Market performance | 119 | 1.65 | 6.89 | 4.880 | 1.922 |
| Product performance | 119 | 1.78 | 6.89 | 4.344 | 1.365 |

Source: Field survey.

Furthermore, the Cronbach $\alpha$ coefficient was used to test the reliability of the questionnaire samples. The results are shown in Table 2. In Table 2, Cronbach $\alpha$ coefficients of enterprise management, knowledge sharing, employee innovation, market performance, and product performance in the questionnaires are 0.934, 0.849, 0.854, 0.911, and 0.886, respectively. Cronbach $\alpha$ coefficients of all items in the questionnaire were greater than 0.7, indicating that the overall reliability of the questionnaire passes the test.

**Table 2.** Reliability test of statistical questionnaire results.

| Indicators | Cronbach $\alpha$ Coefficient | Number of Items |
|---|---|---|
| Enterprise management | 0.934 | 10 |
| Knowledge sharing | 0.849 | 4 |
| Employee innovation | 0.854 | 5 |
| Market performance | 0.911 | 4 |
| Product performance | 0.886 | 8 |

Moreover, the correlation analysis and differential validity results of different questionnaire items were tested, and the results are shown in Table 3. The correlation between enterprise management and knowledge sharing, employee innovation, market performance, and product performance is significant at 0.01. The correlation between employee innovation and market performance, product performance, and other projects is at 0.05. Therefore, there is a significant positive correlation between the variables studied.

**Table 3.** Results of correlation analysis and differential validity of different items in the questionnaire.

| Indicators | Enterprise Management | Knowledge Sharing | Employee Innovation | Market Performance | Product Performance |
|---|---|---|---|---|---|
| Enterprise management | 0.808 | — | — | — | — |
| Knowledge sharing | 0.394 ** | 0.781 | — | — | — |
| Employee innovation | 0.442 ** | 0.435 ** | 0.756 | — | — |
| Market performance | 0.340 ** | 0.354 ** | 0.354 * | 0.853 | — |
| Product performance | 0.168 ** | 0.468 ** | 0.652 * | 0.467 ** | 0.851 |

Note: * represents $p < 0.05$, and ** represents $p < 0.01$.

### 3.2. Empirical Estimation

Evolutionary game theory was used to reflect the premise of bounded rationality and reaching a game equilibrium state in an individual trial-and-error approach. In the green production management and innovation alliance of technology-based SMEs, members lack sufficient trust at the beginning of the partnership due to the diversity and uncertainty of the internal and external environments. With time, the trust between members improves. Consequently, the various alliance systems improve. The optimal strategy for collaborative innovation is finally obtained to achieve the stable operation and development of the partnership after repeated games and constant adjustment of the members' individual choices. The main technical contribution of this study was to use game theory to evaluate the green production management of enterprises and further analyze the evolutionary game process of enterprise production management. Therefore, the method of the evolutionary game was employed to analyze the stability of the innovation alliance of technology-based SMEs under the condition of bounded rationality.

Firstly, it is supposed that the green production management and innovation alliance of technology-based SMEs can be expressed as $A = \{a_1, a_2, \cdots, a_n\}$. Where, $a_i, i = 1, 2, \cdots, n$ refers to the member enterprises of the green production management and innovation alliance. Furthermore, it is assumed that $a_i$ is bounded rationality, the information is completely asymmetric, and there is a tendency to speculate. Secondly, the number of members of the green production management and innovation alliance for different technology-based SMEs is different. Alliance types also vary; therefore, two kinds of coalition members $a_1$ and $a_2$ were selected for evolutionary game analysis without loss of general discussion. The strategy sets of alliance members $a_1$ and $a_2$ are collaborative innovation, independent innovation, and the probability of both parties choosing the "collaborative innovation" strategy is $x_1, x_2 (0 \leq x_1 \leq 1, 0 \leq x_2 \leq 1)$. The possibility of selecting the "independent innovation" strategy is $1 - x_1, 1 - x_2$. Over time, the values of $x_1$ and $x_2$ will change accordingly.

Based on the above basic assumptions, the evolutionary game model of technology-based SMEs can be constructed. The game revenue payment matrix $K$ of collaborative innovation of the innovation alliance of technology-based SMEs can be obtained, as shown in Equation (1):

$$K = \begin{bmatrix} \lambda(C_1 + C_2)p - FC_1 - S, (1 - \lambda)(C_1 + C_2)p - FC_2 - S & R_1 - FC_1 - S, R_2 - fC_2 \\ R_1 - fC_1, R_2 - FC_2 - S & R_1 - fC_1, R_2 - fC_2 \end{bmatrix} \tag{1}$$

$$F = f - q. \tag{2}$$

In Equation (1), the independent innovation resources owned by alliance members $a_1$ and $a_2$ are $C_1$ and $C_2$. When members independently innovate, their incomes are $R_1$ and $R_1$, and they face the same risks. The risk coefficient is $f, 0 \leq f \leq 1$, and the cost of independent innovation by members is $fC_1$ and $fC_2$. In the collaborative innovation process, the dangers faced by alliance members mainly depend on the degree of synergy between the two parties $q, 0 \leq q \leq 1$. The higher the synergy between the two parties, the smaller the risk of collaborative innovation. Therefore, the risk coefficient $F$ of collaborative innovation can be expressed as Equation (2). The cost of collaborative innovation is $FC_1$ and $FC_2$ after members $a_1$ and $a_2$ join the alliance. The organization cost of the alliance is a fixed constant $S$, which refers to the internal and external costs and coordination costs of the normal operation of the alliance. The costs for enterprises $a_1$ and $a_2$ choosing to join the innovation alliance for collaborative innovation are $FC_1 + S$ and $FC_2 + S$. The synergistic utilization effect coefficient of innovative resources is $p$. When member $a_1$ conducts collaborative innovation, the profit distribution ratio is $\lambda, 0 \leq \lambda \leq 1$. Then, the total benefit of collaborative innovation of member $a_1$ is $\lambda(C_1 + C_2)p$. When member $a_2$ conducts collaborative innovation, the profit distribution ratio is $(1 - \lambda)$, and the total use of collaborative innovation of member $a_2$ is $(1 - \lambda)(C_1 + C_2)p$.

According to the payoff matrix $K$ of the game mentioned above, the expected returns of enterprise member $a_1$ when he chooses the strategies of "collaborative innovation" and "independent innovation" are $U_{1t}$ and $U_{1n}$. The average income is $\overline{U}_1$, as shown in Equations (3)–(5):

$$U_{1t} = \lambda x_2(C_1 + C_2)p + R_1 - FC_1 - S - x_2R_1 \tag{3}$$

$$U_{1n} = x_2(R_1 - fC_1) + (1 - x_2)(R_1 - fC_1) = R_1 - fC_1 \tag{4}$$

$$\overline{U}_1 = x_1U_{1t} + (1 - x_1)U_{1n} = x_1x_2[\lambda(C_1 + C_2)p - R_1] + x_1(fC_1 - FC_1 - S) + R_1 - fC_1. \tag{5}$$

The expected benefits obtained by the member $a_2$ when he chooses the strategies of "collaborative innovation" and "independent innovation" are $U_{2t}$ and $U_{2n}$, respectively. The average revenue is $\overline{U}_2$, which is expressed as:

$$U_{2t} = x_1(1 - \lambda)(C_1 + C_2)p + R_2 - FC_2 - S - x_1R_2 \tag{6}$$

$$U_{2n} = x_1(R_2 - fC_2) + (1 - x_1)(R_2 - fC_2) = R_2 - fC_2 \tag{7}$$

$$\overline{U}_2 = x_2U_{2t} + (1 - x_2)U_{2n} = x_1x_2[(1 - \lambda)(C_1 + C_2)p - R_2] + x_2(fC_2 - FC_2 - S) + R_2 - fC_2. \tag{8}$$

According to the replication dynamic equation of the evolutionary game, the replication dynamic equation of alliance member $a_1$ can be expressed as Equation (9):

$$f(x_1) = \frac{dx_1}{dt} = x_1(1 - x_1)\{x_2[\lambda(C_1 + C_2)p - R_1] - (F - f)C_1 - S\}. \tag{9}$$

Similarly, the replication dynamics equation of alliance member $a_2$ can be expressed as Equation (10):

$$f(x_2) = \frac{dx_2}{dt} = x_2(1 - x_2)\{x_1[(1 - \lambda)(C_1 + C_2)p - R_1] - (F - f)C_2 - S\}. \tag{10}$$

The stability of the evolutionary game model of technology-based SMEs was further analyzed. Firstly, the strategy selection of alliance member $a_1$ was analyzed, as shown in Figure 3. In Figure 3, when $(FC_1 + S) < f$, its replication dynamic stability analysis is demonstrated in Figure 3a. Furthermore, $x_2^* < 0$, there is always $x_2 > x_2^*$. At this time, only when $x_1 = 1$ is there $f'(x_1) < 0$. Therefore, $x_1 = 1$ is an evolutionarily stable strategy. When the cost for an alliance member enterprise $a_1$ to choose the "independent innovation"

strategy is greater than the cost of choosing the "collaborative innovation" strategy, the enterprise tends to choose the "collaborative innovation" strategy:

$$f'(x_1) = \frac{df(x_1)}{dx_1} = (1 - 2x_1)[\lambda x_2(C_1 + C_2)p - (F - f)C_1 - S]. \tag{11}$$

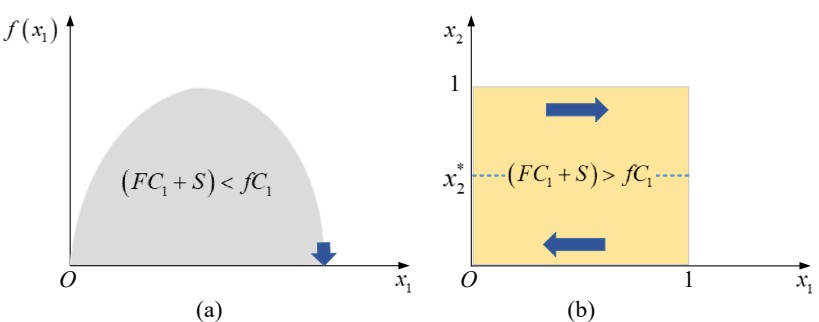

**Figure 3.** (**a**,**b**) Schematic diagram of strategy selection of alliance member $a_1$.

When $(FC_1 + S) > f$, its replication dynamic stability analysis is shown in Figure 3b. $x_2^* < 1$ can be obtained from the model. At this time, it can be divided into two cases for discussion. When $x_2 > x_2^*$, $f(x_1) > 0$, $f'(1) < 0$, and $f'(0) > 0$, $x = 1$ is the stable point of evolution. The enterprise $a_1$ will finally choose the "collaborative innovation" strategy. When $x_2 < x_2^*$, $f(x_1) < 0$, $f'(1) > 0$, and $f'(0) < 0$, $x = 0$ is the stable point of evolution. The enterprise $a_1$ will finally choose the "independent innovation" strategy.

Similarly, for member enterprises $a_2$, the evolutionary game process is similar to $a_1$. In the case of $x_1 = x_1^*$, $x_1 > x_1^*$, $x_1 < x_1^*$, its game stability points are any $x_2$, one, and zero, respectively, as shown in Figure 4.

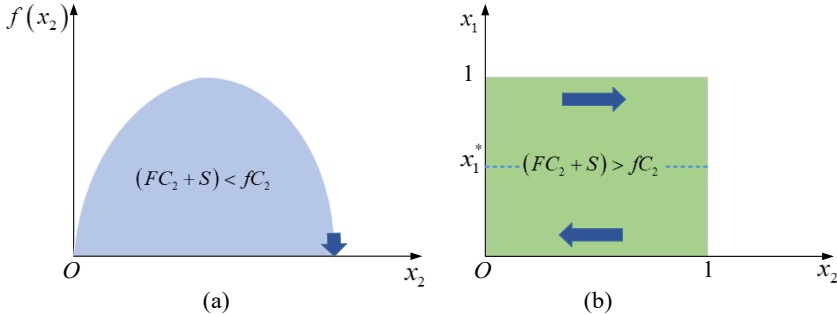

**Figure 4.** (**a**,**b**) Schematic diagram of strategy selection of alliance member $a_2$.

## 4. Results

### 4.1. Results of Green Production Management and Innovation Based on Product Cycle and Enterprise Visits

The results of green production management and innovation of technology-based SMEs are analyzed and shown in Figure 5. In Figure 5, among the 119 companies surveyed, 14% are in the introduction stage, 69% are in the growth stage, and 16% are in the mature stage. It was found that most of the technology-based SMEs surveyed are in the growth stage, and the enterprises are still vigorous. In the figure, the number of page views of SMEs in different periods is different. The page views in the introduction period are 126, in the growth period are 278, in the mature period are 354, and in the decline period are 318.

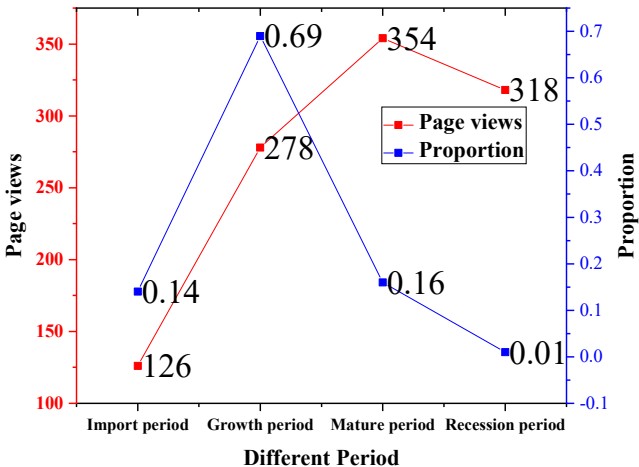

**Figure 5.** Product life cycle and enterprise visits.

*4.2. Results of Green Production Management and Innovation Based on Core Elements*

The results of green production management and innovation based on core elements are shown in Figure 6. In Figure 6, when technology-based SMEs manage and innovate green production, the top six core influencing factors include innovation projects, A1(32.19%); scientific researchers, A2(25.48%); high-end technology or patents, A3(24.43%); fresh graduates, A4(16.18%); ordinary people, A5(14.08%); incubation platform incubation, A6(11.21%); and others, A7(5.14%). Innovative projects, scientific researchers, and high-end technologies or patents are the main source factors for green production management and innovation in the establishment of technology-based SMEs. Fresh graduates include all college students who graduated throughout the year. The total number and scale of fresh graduates determine the efficiency of enterprise production management innovation. Access duration refers to the amount of time the system spends on page visits during data processing. As seen in Figure 6, the average visit time of A2, A4, and A6 is long. The average access time of A2 and A4 is more than 35s, and the average access time of A1 and A7 is short. Overall, A1 has the shortest average visit time, while A4 has the longest average visit time.

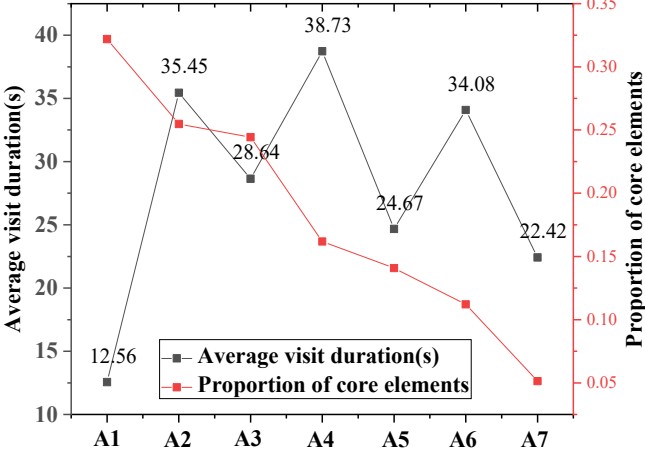

**Figure 6.** Achievements of green production management and innovation core elements.

*4.3. Results of Green Production Management and Innovation within Enterprises*

The analysis of the enterprise's internal green production management and innovation is portrayed in Figure 7, which shows that 51 major enterprises transform their technologies through independent development in enterprise technology transformation methods. In addition, 36 companies changed their technologies through cooperative development, and

15 companies transformed their technologies through authorization. Only eight companies have converted their technologies through transfer. The results of the above data indicate that the technology-based SMEs in Zhejiang Province mainly transform their technologies through independent and cooperative development when they conduct internal green production management and innovation. Further analysis of obstacles shows that the lack of scientific and technological talents and insufficient research funds are the obstacles facing most technology-based SMEs to carry out internal green production management and innovation. Miao et al. [19] proposed that collaborative innovation should be strengthened. Talent is the most difficult heterogeneous asset to navigate and define. It is necessary to boldly implement a high-level talent mutual recruitment plan between industries, universities, and research institutions and activate intellectual property rights development. Cooperation between enterprises, universities, and research institutes must be strengthened via technical research cooperation, sharing scientific and technological equipment, theoretical research, and other diversified cooperation methods. The government plays a leading role in promoting and regulating the effectiveness of resource allocation.

**(a)**

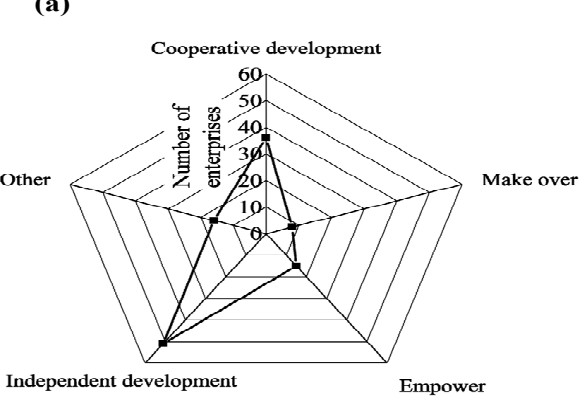

**(b)**

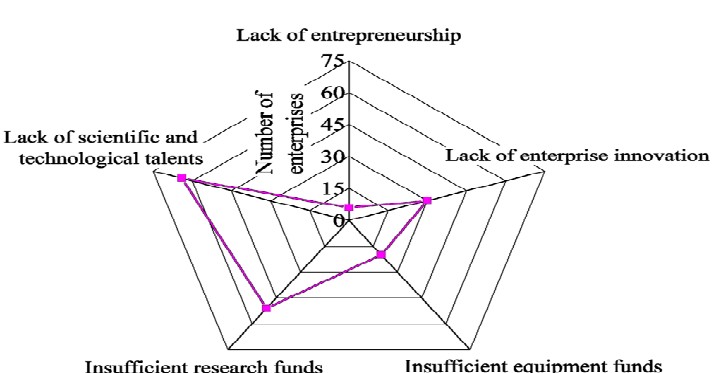

**Figure 7.** The results of green production management and innovation within the enterprise; (**a**) The way of enterprise technology transformation (**b**) The main obstacles to innovation within the enterprise.

Moreover, innovation support services should be improved. Government departments and public service platforms of science and technology should provide comprehensive and full-featured services for technology-based SMEs and increase financial institutions and venture capital institutions to support technology-based SMEs. Furthermore, government departments vigorously develop strategic emerging technology companies. For some high-tech products with good prospects, government departments help them expand the market and find a docking application market [20]. Additionally, this paper proposes that the equilibrium scale of the system should be grasped. The technology-based SMEs

innovation ecosystem should change the technology resource transaction mode among the collaborative innovation subjects and adopt a hybrid implementation mode of technology resource transaction. When the innovation ecosystem of technology-based SMEs reaches a certain scale, they should grasp the system's balanced scale, clarify the innovation's main direction, and form a relatively stable network structure. In addition, attaining the appropriate scale of the innovation system will help improve collaborative innovation's overall effect. Lastly, one should consider improving the innovation environment. Science and technology industrial parks should try their best to play the role of innovation carriers, strive to create an innovation culture in the gardens, and meet the requirements of proper planning and complete supporting services [21,22]. In the science and technology industrial park, technology-based SMEs can fully use the advantages of personnel, technology, and capital. Following their successful integration, the innovation ecosystem of technology-based SMEs circulates in a good state.

### 4.4. Results of Green Production Management and Innovation Based on Support by Government Agencies

The results of green production management and innovation based on support from government agencies are portrayed in Figure 8. From Figure 8, it is apparent that the score of product transformation that is not satisfied with enterprise management is 48, while the score of product transformation that is very satisfied with enterprise management is only 42. In addition, most people rate the market performance of enterprise products as very satisfactory. In the score proportion of employee innovation projects, the difference between the scores of dissatisfied and very dissatisfied is small, indicating that most employees hope the enterprise will strengthen innovation support for green production management.

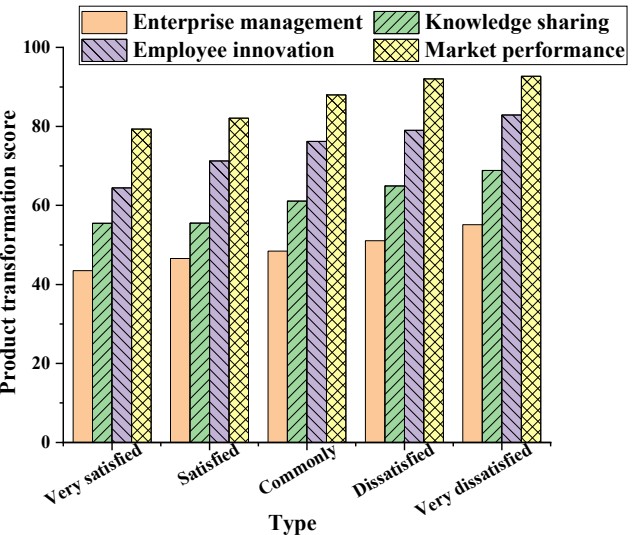

**Figure 8.** Results of green production management and innovation supported by government agencies.

### 4.5. Results of Environmental Factors Restricting Green Production Management and Innovation

For obvious illustration, the factors restricting technology-based SMEs' green production management and ecological innovation environment were analyzed, as shown in Figure 9. The results show that the innovation environment has a greater impact on product search, followed by the talent environment. This suggests that the market environment is the primary constraint, and the infrastructure environment is the least significant constraint. Therefore, the innovation of production management of enterprises needs to strengthen infrastructure construction in the financial climate.

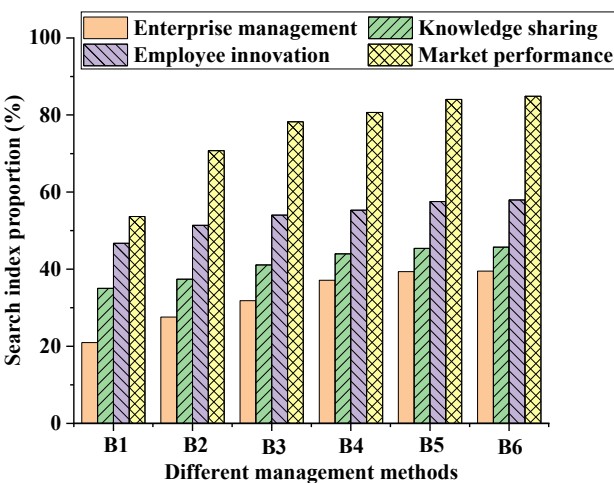

**Figure 9.** Results of environmental factors restricting green production management and innovation of enterprises (B1: Market environment; B2: Technical environment; B3: Resource environment; B4: Infrastructure environment; B5: Financial background; B6: Innovation environment).

## 5. Conclusions

This paper is based on literature research, social survey, and case analysis. Digital psychology and the evolutionary game model are integrated to design and improve the structural model and operating mechanism of the green ecology and innovative production system of technology-based SMEs. Based on the empirical results mentioned above, we found that the innovation of green production management and innovation ecosystem construction of technology-based SMEs faces the following challenges. Firstly, innovative development is still facing certain difficulties. Secondly, many technology-based SMEs do not realize the importance of open, collaborative innovation. Thirdly, it is difficult for technology-based SMEs to cooperate in creation. Fourthly, the government, financial institutions, and social organization service support systems are imperfect. Therefore, this paper proposes that the green production management and innovation ecosystem construction of the innovation of technology-based SMEs should take necessary and potential strategies to overcome the glitches they face. The main mechanism for the formation of relevant conclusions is to adopt the relevant management strategies of the green production management model and further analyze the challenges faced by the innovation ecosystem. It can be found that the reason for the emergence of related challenges is that enterprises have certain loopholes in their production management, and the management mode and enterprise production efficiency are not matched. In addition, the main contribution of this paper to scientific knowledge is mainly reflected in the optimization of psychological models and the profound reform of management mechanisms, which has practical application value for the technical composition of the industrial chain of SMEs.

So, based on the above findings, this paper put forward experimental directions for the construction and healthy operation of the innovation ecosystem of follow-up technology-based enterprises. Some shortcomings, such as issues related to the structure of the green production management and innovation ecosystem operating mechanism, co-evolutionary game mechanism, and crisscrossing complex relationships of technology-based SMEs, have not been further studied. In future research, it is necessary to combine the correlation between green production management and the innovation ecosystem of technology-based SMEs to further improve the efficiency of enterprise production management.

**Author Contributions:** Conceptualization, L.Z. and Y.D.; methodology, Y.D.; formal analysis, Y.D.; resources, L.Z.; data curation, Y.D.; writing—original draft preparation, L.Z.; writing—review and editing, Y.D. All authors have read and agreed to the published version of the manuscript.

**Funding:** This research received no external funding.

**Institutional Review Board Statement:** Not applicable.

**Informed Consent Statement:** Informed consent was obtained from all subjects involved in the study.

**Data Availability Statement:** The study will be available on request from authors.

**Conflicts of Interest:** The authors declare no conflict of interest.

## Appendix A

**Table A1.** Study questionnaire structure.

| Questionnaire Items | Specific Questions |
|---|---|
| The basic situation of the enterprise | The total number of employees? A. Less than 500 B. 500~2000 C. 2000 or more |
| | Total business assets? A. Less than 500,000 B. 500,000~2 million C. 2 million or more |
| Management | Do you understand the internal management mechanism of the enterprise? A. Not knowing much B. Know a little C. Know very well |
| Knowledge sharing | How much support is there for knowledge-sharing platforms within the enterprise? A. Not very clear B. Know a little C. Very supportive |
| Employee innovation | How much does the enterprise reward employees for innovative behavior? A. Not very clear B. General C. The reward is strong |
| Market performance | How are enterprise products currently performing in the market? A. Not very clear B. General C. Very well |
| Product performance | What is the performance of the products produced by the enterprise? A. Not very clear B. General C. Very good performance |

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
