# Peer review of "Green Production Management and Innovation Nexus: Evidence from Technology-Based SMEs of China"

_sustainability, doi:10.3390/su15064710_

Round 1
Reviewer 1 Report
The article deals with the current topic and appropriate data processing methods are used in the paper.
However, the following recommendation ought to be incorporated:
The aim and benefits of the article is necessary to clearly state in abstract and introduction part.
In the theoretical part it would be beneficial to better elaborate what part of scientific knowledge the article adds.
The paper should include more literature sources to describe and expand the theoretical background in more detail. In the theoretical section, it would be useful to clearly state what research gap the article complements.
The conclusion should further articulate the article's contribution to scientific knowledge and articulate future research in more detail.
Author Response
Reviewer 1:
Comments and Suggestions for Authors
The article deals with the current topic and appropriate data processing methods are used in the paper.
However, the following recommendation ought to be incorporated:
The aim and benefits of the article is necessary to clearly state in abstract and introduction part.
Reply: Thank you for your valuable advice. The purpose of the manuscript is to study the relationship between green production management and enterprise innovation. The benefit of research is that it can promote the efficiency of enterprise production management. The purpose and benefits of this research have been explained in detail in the abstract and introduction.
In the theoretical part it would be beneficial to better elaborate what part of scientific knowledge the article adds.
Reply: Thank you for your valuable advice. We added the main scientific knowledge about green production and enterprise management system innovation. The practical benefits of system innovation of green production management have been supplemented in detail in Section 2.1.
The paper should include more literature sources to describe and expand the theoretical background in more detail. In the theoretical section, it would be useful to clearly state what research gap the article complements.
Reply: Thank you for your valuable advice. By expanding the theoretical knowledge of production management of small and medium-sized scientific and technological enterprises, the research has filled the research gap in production management of scientific and technological enterprises. The expansion of the theoretical background of the research has been supplemented in detail in the introduction.
The conclusion should further articulate the article's contribution to scientific knowledge and articulate future research in more detail.
Reply: Thank you for your valuable advice. The main contribution of this article to scientific knowledge is mainly reflected in the optimization of psychological model and the profound change of management mechanism, which has practical application value for the technical composition of SME industrial chain. The main contribution of the article and the main suggestions for future research has been supplemented in detail in the conclusion.

Reviewer 2 Report
Referee’s report
Sustainability
Manuscript Number: sustainability-2179381
Title: Green production management and innovation nexus: evidence from technology-based SMEs of China
This study designs and improves the structural and operating mechanism of technology-based SMEs by employing the person fit and evolutionary game models, and gathers data from technology-based SMEs of Zhejiang Province, China, by conducting a questionnaire-based survey. They find that the principle of person-environment fit revealed the positive leadership skills of enterprise managers; the evolutionary game model revealed the re-optimization of SMEs to improve management efficiency through reforming enterprises' organization, management and supervision mechanism.
This paper applies unique data and the modern empirical method, and presents some unique insights. The findings of this paper are also interesting.
However, some of these points have been identified to strengthen the study:
Major points
1. Elements that should be covered in the Introduction:
o State the research problem (purpose of the study).
o State the aims of the study. The following is a list of questions:
Is there a problem? Why is does it exist? Why does it need to be solved?
Who will benefit from the study? In what sense will they benefit?
How will it contribute to what is already known?
An important part of introduction is where you state the proposal objectives. (After addressing the above questions).
o Provide the context and set the stage for the research question and show its necessity and importance.
2. The author's contribution to the empirical model in Section 3.2 should be made clear, and if there is no specific contribution, this section should be appropriately concise.
3. The authors find that in conclusion “Based on the empirical results mentioned above, it is found that the innovation of green production management and innovation ecosystem construction of technology-based SMEs confront the following challenges.” However, this paper does not analyze the mechanism and reason of the relevant conclusion.
4. Expressions and Copy Editing: There were some non-standard expressions, and typos.

Author Response
Reviewer 2:
This study designs and improves the structural and operating mechanism of technology-based SMEs by employing the person fit and evolutionary game models, and gathers data from technology-based SMEs of Zhejiang Province, China, by conducting a questionnaire-based survey. They find that the principle of person-environment fit revealed the positive leadership skills of enterprise managers; the evolutionary game model revealed the re-optimization of SMEs to improve management efficiency through reforming enterprises' organization, management and supervision mechanism.
This paper applies unique data and the modern empirical method, and presents some unique insights. The findings of this paper are also interesting.
However, some of these points have been identified to strengthen the study:
Major points
- Elements that should be covered in the Introduction:
o State the research problem (purpose of the study).
Reply: Thank you for your valuable advice. The purpose of the study is to optimize the green management and production innovation strategies of small and medium-sized science and technology enterprises, and further improve the cooperative efficiency of enterprise management. The specific purpose and problems of the research have been supplemented in detail in the introduction.
o State the aims of the study. The following is a list of questions:
Is there a problem? Why is does it exist? Why does it need to be solved?
Who will benefit from the study? In what sense will they benefit?
How will it contribute to what is already known?
An important part of introduction is where you state the proposal objectives. (After addressing the above questions).
Reply: Thank you for your valuable advice. The main problem of the current research is that the efficiency of enterprise production management is not matched with the management mechanism. The contribution is to realize the innovation of green production management mode through institutional mechanism reform. The significance of the study is to combine the new production strategy of small and medium-sized enterprises and further promote the improvement of enterprise production management efficiency. The explanation of the research question has been supplemented in detail in the fourth paragraph of the introduction section.
o Provide the context and set the stage for the research question and show its necessity and importance.
Reply: Thank you for your valuable advice. The background of the study is weak economic growth and weak enterprise vitality. The necessity and importance of the economic system reform of small and medium-sized enterprises have been supplemented in detail in the introduction section.
- The author's contribution to the empirical model in Section 3.2 should be made clear, and if there is no specific contribution, this section should be appropriately concise.
Reply: Thank you for your valuable advice. The main technical contribution of the research is to use game theory to evaluate the green production management of enterprises. The specific contribution to the empirical model has been supplemented in detail in Section 3.2.
- The authors find that in conclusion “Based on the empirical results mentioned above, it is found that the innovation of green production management and innovation ecosystem construction of technology-based SMEs confront the following challenges.” However, this paper does not analyze the mechanism and reason of the relevant conclusion.
Reply: Thank you for your valuable comments. The main mechanism for the formation of relevant conclusions is to adopt relevant management strategies of green production management mode. The main reason is that enterprises have certain loopholes in their own production management, and the management mode does not match the production efficiency of enterprises. The mechanism and reasons for the formation of relevant conclusions have been supplemented in the conclusion section.
- Expressions and Copy Editing: There were some non-standard expressions, and typos.
Reply: Thank you for your valuable comments. The article has been carefully checked by professional editors, and all the typos have been revised.

Reviewer 3 Report
This is an interesting piece of empirical work. The theory discussion is also quite nice. I noticed the citations - nothing older than 2019 which seems odd. Nothing relevant to this paper before then?
The setting/context is interesting. Smaller firms in China, technology and green manufacturing.
I was surprised that the survey questions were not included in the paper (appendix?)or a link to find them. It can be hard to judge things if we do do not see them.
Looking at independent innovation vs. collaborative innovation is valuable, so the modelling was interesting.
"fresh graduates" is important, but I wondered was this in reference to anything more specific? type of graduates, quantity, quality/attributes?
page views was mentioned in Figure 5 but nowhere else in the article.
Figure 6 has visit duration but it is not explained in the article.
Author Response
Reviewer 3:
Comments and Suggestions for Authors
This is an interesting piece of empirical work. The theory discussion is also quite nice. I noticed the citations - nothing older than 2019 which seems odd. Nothing relevant to this paper before then?
Reply: Thank you for your comments. Relevant documents before 2019 have been added to the article. The number of the documents in the article is [2] - [4]. These documents are about enterprise system innovation. Thank you again for your valuable comments.
The setting/context is interesting. Smaller firms in China, technology and green manufacturing.
Reply: Thank you for your valuable comments. The research background is the green production management and economic system innovation of small and medium-sized enterprises in China, which has been supplemented in the first paragraph of the introductionsection.
I was surprised that the survey questions were not included in the paper (appendix?)or a link to find them. It can be hard to judge things if we do do not see them.
Reply: Thank you for your valuable comments. The questions about the questionnaire survey have been supplemented in detail in Table 1. Thank you again for your comments.
Looking at independent innovation vs. collaborative innovation is valuable, so the modelling was interesting.
Reply: Thank you for your recognition of the research content of the article, and express your heartfelt thanks for your contribution to the article review.
"fresh graduates" is important, but I wondered was this in reference to anything more specific? type of graduates, quantity, quality/attributes?
Reply: Thank you for your valuable advice. "Fresh graduates" refers to all college students who graduated in that year. The relevant contents about the types and quantity of graduates have been supplemented in detail in Section 4.2.
page views was mentioned in Figure 5 but nowhere else in the article.
Reply: Thank you for your opinion. The number of page views in the introduction period is 126, the growth period is 278, the maturity period is 354, and the decline period is 318. The specific information about page views has been explained above Figure 5.
Figure 6 has visit duration but it is not explained in the article.
Reply: Thank you for your opinion. The average visit time of A2, A4 and A6 is long, among which the average visit time of A2 and A4 is over 35s, while that of A1 and A7 is short. The detailed explanation of the access duration has been supplemented in detail above Figure 6.

Round 2
Reviewer 1 Report
The empirical article was improved by the authors according to the requirements and recommendations.
Reviewer 2 Report
The analysis does, in my view, meet the standards required for articles published in Sustainability.